# Causes and Pathophysiology of Acquired Sideroblastic Anemia

**DOI:** 10.3390/genes13091562

**Published:** 2022-08-30

**Authors:** Juan Jose Rodriguez-Sevilla, Xavier Calvo, Leonor Arenillas

**Affiliations:** 1Department of Hematology, Hospital del Mar, 08003 Barcelona, Spain; 2Laboratori de Citologia Hematològica, Department of Pathology, Hospital del Mar, 08003 Barcelona, Spain; 3Group of Translational Research on Hematological Neoplasms (GRETNHE), IMIM-Hospital del Mar, 08003 Barcelona, Spain

**Keywords:** sideroblastic anemia, ring sideroblast, MDS, MDS/MPN-RS-T, *SF3B1*

## Abstract

The sideroblastic anemias are a heterogeneous group of inherited and acquired disorders characterized by anemia and the presence of ring sideroblasts in the bone marrow. Ring sideroblasts are abnormal erythroblasts with iron-loaded mitochondria that are visualized by Prussian blue staining as a perinuclear ring of green-blue granules. The mechanisms that lead to the ring sideroblast formation are heterogeneous, but in all of them, there is an abnormal deposition of iron in the mitochondria of erythroblasts. Congenital sideroblastic anemias include nonsyndromic and syndromic disorders. Acquired sideroblastic anemias include conditions that range from clonal disorders (myeloid neoplasms as myelodysplastic syndromes and myelodysplastic/myeloproliferative neoplasms with ring sideroblasts) to toxic or metabolic reversible sideroblastic anemia. In the last 30 years, due to the advances in genomic techniques, a deep knowledge of the pathophysiological mechanisms has been accomplished and the bases for possible targeted treatments have been established. The distinction between the different forms of sideroblastic anemia is based on the study of the characteristics of the anemia, age of diagnosis, clinical manifestations, and the performance of laboratory analysis involving genetic testing in many cases. This review focuses on the differential diagnosis of acquired disorders associated with ring sideroblasts.

## 1. Introduction

Ring sideroblasts are erythroblasts with an abnormal accumulation of iron in their perinuclear mitochondria, and their presence defines sideroblastic anemias. To reveal them, it is necessary to apply Prussian blue stain (Perls’ reaction) to bone marrow aspirate smears. Ring sideroblasts are found in a variety of pathological conditions, both congenital and acquired. Among the acquired causes of sideroblastic anemias, we can find clonal and non-clonal disorders.

The mechanisms that lead to the different sideroblastic anemias are heterogeneous, but in all of them, the abnormal deposition of iron is due to disturbances of mitochondrial proteins regulating Heme synthesis or Fe/S cluster synthesis, as well as translation impairment of mitochondrially encoded proteins. As a consequence of these alterations, ineffective erythropoiesis and tissue iron overload emerge. In the last three decades, a deep understanding of the pathophysiological mechanism has been accomplished and the bases for possible targeted treatments have been established [1]. This review focuses on the differential diagnosis of acquired disorders associated with ring sideroblasts.

## 2. Historical Context

In 1942, Hans Grüneberg demonstrated, using the Prussian blue staining, the presence of free iron in the cytoplasm of some erythroblasts (sideroblasts) and in some mature erythrocytes (siderocytes) [2]. In 1945, Cooley [3] described a patient with sex-linked anemia, which probably corresponded to a case of the nonsyndromic form of X-linked SA (XLSA), since later Cotter et al. identified mutations in *ALAS2* and ringed sideroblasts in the same family [4]. In 1956, Björkman described a series of four patients with chronic refractory anemia with numerous abnormal bone marrow sideroblasts, one of which developed leukemia. This is probably the first description of myelodysplastic syndromes with ring sideroblasts [5]. Sideroblastic anemias were recognized as a specific subtype of anemia in the 1960s [6]. Within the last 30 years, with the important advances in molecular biology, the genetic origin of more than two-thirds of congenital sideroblastic anemias cases, and a great proportion of cases of acquired clonal disease have been clarified.

## 3. Ring Sideroblast Definition

Prussian blue staining (Perls’ reaction) is an essential technique in the study of patients with anemia. Its application in bone marrow aspirate smears permits the analysis of macrophage iron storage and the assessment of the number and characteristics of sideroblasts. This stain reveals ferritin granules within the erythroblasts and hemosiderin in bone marrow macrophages [7]. 

A proportion of normal erythroblasts exhibit few (1–5) iron-containing granules randomly distributed around the cytoplasm. Such erythroblasts are designated as sideroblasts and stand for 20–50% of the erythroblasts in a normal bone marrow [8]. Sideroblasts are visible in bone marrow aspirate smears but not in bone marrow biopsy sections since erythroblastic iron is lost in bone marrow biopsy processing [7]. Ring sideroblasts are aberrant sideroblasts where iron-loaded mitochondria are visualized by Prussian blue staining as a perinuclear ring of green-blue granules. There have been various definitions of a ring sideroblast causing confusion and controversy among clinicians [9]. The International Working Group on Morphology of Myelodysplastic Syndrome (IWGM-MDS) defined three types of sideroblasts [10]:-Type 1 sideroblasts: < 5 siderotic granules in the cytoplasm.-Type 2 sideroblasts: ≥ 5 siderotic granules, but no perinuclear distribution.-Type 3 or ring sideroblasts: ≥ 5 siderotic granules in a perinuclear position, covering at least one-third of the nuclear circumference.

To establish the percentage of sideroblasts in a bone marrow, at least 100 erythroid precursors of all maturation stages should be counted. This definition of ring sideroblast proposed by the IWG-MDS was incorporated into the 2008 and updated in the 2017 edition of the WHO classifications of Tumours of Haematopoietic and Lymphoid Tissues [7,11].

## 4. Classification of Sideroblastic Anemias

Sideroblastic anemias can be divided into congenital sideroblastic anemias and acquired forms. Congenital sideroblastic anemias incorporate nonsyndromic and syndromic conditions. Acquired sideroblastic anemias include etiologies that range from clonal disorders (e.g., MDS with ring sideroblasts and MDS/MPN neoplasm with ring sideroblasts and thrombocytosis) to toxic or metabolic acquired sideroblastic anemia. 

Table 1 shows classification, genetic, and clinical features of the sideroblastic anemias.

## 5. Ring Sideroblast Formation

The discovery of genetic variations underlying ring sideroblast has led to a better understanding of the pathophysiology of the sideroblastic anemias. Nevertheless, our understanding of how ring sideroblasts arise is limited. There are many open questions in this regard: Are they detrimental to the erythroblast? Are they a cause or a consequence?

Mitochondrion is the epicenter of sideroblastic anemia. Disrupted mitochondrial metabolism is present among all sideroblastic anemias for which an etiology has been defined. The mitochondrial functions affected in sideroblastic anemias are Heme biosynthesis; iron–sulfur cluster (ISC) biogenesis; and synthesis of mitochondrial proteins, general proteins, or proteins dedicated to oxidative metabolism. All of these defects lead to an aberrant accumulation of iron in the mitochondria of erythroblasts [1,12]. Figure 1 represents the main causes of acquired sideroblastic anemia.

Mitochondria provide the majority of the ATP needed by eukaryotic cells through oxidative phosphorylation [13]. The adult erythrocyte is the only mammalian cell that does not have mitochondria, relying exclusively on anaerobic glycolysis for ATP production [14]. Mitochondria are semi-autonomous organelles that most likely evolved from free-floating prokaryotes that infiltrated eukaryotic cells over a billion years ago [15]. 

The mitochondria genome is small, around 16 kb, and replicates autonomously conserving vestiges of their prior self-sufficiency [16,17]. Mitochondrial DNA, along with several bacterial genomes, displays an intron-free circular structure [18]. Chromatin absence and a limited DNA-repair capacity enable mutations in the mitochondrial DNA to develop sideroblastic anemia [19].

Replication within mitochondria occurs independently of the nuclear genome [20]. Mitochondria are stochastically distributed to progeny after cells undergo mitosis. As a result, acquired mitochondrial abnormalities are passed on in an unequal manner to the daughter cells. This feature is important to some of the hereditary mitochondrial disorders that produce sideroblastic anemia. This characteristic also presents a conundrum regarding acquired sideroblastic anemias. Some cases of sideroblastic anemia linked with myelodysplasia include mutations that prevent some cytochromes from working properly [21,22]. It remains uncertain how mitochondria with deteriorated enzymatic performance become so prevalent in cells. Reasonably, impairment of the mitochondrion should not confer a survival advantage. 

### 5.1. Heme Synthesis 

Most of congenital sideroblastic anemias are due to Heme deficiency. Heme is a critical component of several mitochondrial enzymes (cytochromes b, c1, c, a, and a3), as well as cytosolic enzymes such as catalase [23]. Heme plays structural and functional roles as an essential member in the hemoglobin structure. Particularly, Heme regulates the translation of globin mRNA, mediates reversible oxygen binding [24], and stabilizes the globin protein chains.

Heme biosynthesis initiates with the condensation of glycine and succinyl-CoA to generate 5′-aminolevulinic acid (ALA) [25], consuming pyridoxal phosphate (active form of vitamin B6) as a cofactor in the reaction [26]. ALA then is transported to cytoplasm, where, after numerous enzymatic reactions, it is converted to coproporphyrinogen III [27]. This molecule again reaches the mitochondrion, where it undergoes further modifications and has iron inserted into the protoporphyrin IX ring by ferrochelatase (FECH), eventually generating Heme [28]. Porphyria is caused by defects in the cytoplasmic phases of Heme production. For instance, functional anomalies of the enzyme porphobilinogen deaminase produce acute intermittent porphyria [29]. Only 10 patients with erythropoietic protoporphyria (EPP) [30], a disorder characterized by pronounced deficiency of FECH, have been reported to present ring sideroblasts [31].

Aminolevulinic acid synthase (ALAS) is both the first and the rate-limiting enzyme in Heme biosynthesis [25]. Heme regulates its activity by inhibiting feedback. The two ALAS genes have been cloned and allocated to specific chromosomal regions. The *ALAS-1* (also known as *ALAS-n*) gene has been localized to chromosome 3 (3p21) [32], being highly expressed in the liver. ALAS-1 maintains steady levels, providing basal Heme production required by all cells. ALAS-1 is a key member in the Heme biosynthetic process in mammalian cells, with the exception of erythroid cells, where erythroid-specific 5-aminolevulinate synthase (ALAS-2 or ALAS-E) governs the initial stage of Heme biosynthesis [33]. This enzyme is encoded by a gene on the X chromosome (Xp11.21), and its expression is restricted to the erythroid lineage [34]. Expression and activity of ALAS-2 is regulated by iron levels, as well as Heme-mediated feedback regulation [35]. Importantly, deficiency of ALAS-2 accounts for around 40% of all congenital sideroblastic anemia cases [36]. 

### 5.2. ISC Biogenesis

Iron–sulfur clusters (ISCs) are core components of many mitochondrial and extramitochondrial proteins, showing catalytic activity [37]. ISC plays a fundamental role in cellular iron uptake regulation, iron storage, Heme synthesis, and interaction with iron regulatory protein 1 (IRP1) and FECH [38]. Congenital sideroblastic anemia, accompanied by defects in the transfer stage of ISC biogenesis, has been reported.

### 5.3. Mitochondrial Respiratory Complex Proteins and Mitochondrial Protein Synthesis 

A broad defect in mitochondrial protein synthesis has been described to lead to congenital sideroblastic anemias associated with neuromuscular disease and lactic acidosis consequent to impaired mitochondrial energy metabolism.

The homeostasis of iron is vital to human health, and iron dyshomeostasis can lead to various disorders since excess iron can promote the generation of deleterious reactive oxygen species (ROS).

Iron homeostasis is maintained by iron regulatory proteins (IRP1 and IRP2) and the iron-responsive element (IRE) signaling pathway [39]. 

Intracellular iron is used for multiple functions; if not utilized, it is stored in ferritin or exported by ferroportin in order to maintain the labile iron pool within narrow limits to avoid toxicity. Erythroblasts are cells specialized in iron uptake, and more than 80% of this iron is directed to mitochondria [40,41].

While normal erythroblasts store their iron in cytosolic ferritin, which is encoded by the *FTH1* and *FTL* genes, ring sideroblasts store their iron in mitochondrial ferritin (FtMt), which is encoded by the FTMT gene, an intronless gene located on chromosome 5q23.1 [42]. FtMt contains ferroxidase activity; thus, it is likely to sequester potentially damaging free iron [43]. Ultimately, overexpression of FtMt results in mitochondrial iron loading and cytosolic iron deficiency [44].

The nature of this iron and the fact that these cells survive this massive overload has long been a conundrum. Bessis and Breton-Gorius found by electron microscopy that this electron dense iron gave images similar to those of the iron cores of ferritin and proposed that it was ferritin [45]. At that time, the structural complexity of ferritin was not known, and there was then no molecular basis for mitochondrial targeting. Different studies showed that there is little, if any, FtMt in normal erythroblasts but very high levels in the iron-loaded mitochondria in ringed sideroblasts [46,47]. 

Through Fenton chemistry (Equation (1)), iron catalyzes the creation of reactive oxygen species [48]. Molecules such as the hydroxyl radical (−OH) form in environments where oxidation processes take place around iron [49]. The mitochondrion’s oxidative metabolic machinery facilitates a suitable setting to produce reactive oxygen species. In sideroblastic anemia, the main damage that results in iron-laden mitochondria might trigger a feedback cycle of aggravating mitochondrial impairment [50]. For instance, (−OH) stimulates the peroxidation of lipids and proteins, as well as the formation of cross-links in DNA strands. Given the previously suggested lack of DNA-repair enzymes in mitochondria, the latter event might be extremely harmful. 

Equation (1) is the Fenton reaction. The Fenton reaction involves iron II (Fe^2+^) reacting with H_2_O_2_ to yield a hydroxy radical (OH) and a hydroxide ion (OH^−^):(1)$${\mathrm{Fe}}^{2+}+H_{2}O_{2}\to {\mathrm{Fe}}^{3+}+{\mathrm{OH}}^{.}+{\mathrm{OH}}^{-}$$

## 6. Diagnosis of Sideroblastic Anemias

Sideroblastic anemia is primarily a laboratory diagnosis, based on the identification of ring sideroblasts in the bone marrow aspirate smear stained with the Perls’ reaction.

The distinction between the different forms of sideroblastic anemia is based on the study of the characteristics of the anemia, age of diagnosis, clinical symptoms (search for symptoms suggestive of syndromic disease), and the performance of laboratory analysis, which, in many cases, involves genetic testing. Congenital sideroblastic anemias’ forms are usually diagnosed in childhood or youth and acquired forms in the elderly. However, some of the congenital sideroblastic anemias show variable expression and may be diagnosed in adulthood [51].

A complete blood count (CBC), peripheral blood smear, comprehensive iron profile (e.g., ferritin, transferrin, and total iron-binding capacity (TIBC), and bonemarrow bone marrow aspiration are some of the indispensable tests that should be performed along evaluation. On the CBC, white blood cell and platelet counts are usually normal; low levels could indicate the presence of splenomegaly/hypersplenism or possible underlying causes, such as myelodysplastic syndrome (MDS). The platelet count is elevated in myeloproliferative/myelodysplastic neoplasms with ring sideroblasts and thrombocytosis (NMP/MDS-RS-T). The hemoglobin level varies between the different types of sideroblastic anemias: in inherited forms, hemoglobin tends to remain stable for long periods of time, and in MDS-RS anemia, it can be slowly progressive [51]. The mean corpuscular volume (MCV) can be a useful tool in distinguishing between the different sideroblastic anemias; most congenital sideroblastic anemias are microcytic, unlike myelodysplastic syndromes with ring sideroblasts, which usually present with macrocytic anemia [52]. The reticulocyte count is usually normal or low, which translates into ineffective erythropoiesis present in most cases.

Sideroblastic anemias are characterized by a variable degree of systemic iron overload more prominently in congenital sideroblastic anemias, carrying significant morbidity and mortality. Mild-to-moderate hepatosplenomegaly is frequently seen, usually with preserved liver function. This iron overload is due to ineffective erythropoiesis, similar to what occurs in congenital dyserythropoietic anemias, thalassemia, and anemias with decreased hepcidin and increased intestinal iron absorption [51,53] In most patients with congenital sideroblastic anemias and with MDS-RS, the study of iron parameters reveals an increase in serum iron, ferritin, and transferrin saturation at diagnosis, even before the patient has required transfusion support [54].

Cytological evaluation of panoptic stain shows red cells with marked anisocytosis and poikilocytosis [7]. Siderocytes and red blood cells (RBCs) in which an anomalous distribution of hemoglobin and basophilic stippling coexist are usually observed [55]. Hemosiderin particles are sometimes large and may be visible with panoptic staining (Pappenheimer bodies) [56].

In the morphological study of the bone marrow aspirate, panoptic staining shows an increase in the erythroid series consisting of erythroblasts with poorly hemoglobinized cytoplasm and basophilic stippling [57]. Signs of dyserythropoiesis, such as megaloblastic changes and multinuclearity, are also seen in MDS-RS [10]. Cytoplasmic vacuolization of myeloid precursors and immature erythroid forms is common in Pearson’s syndrome [58], MLASA, and copper deficiency [8,9,10].

To reveal ring sideroblasts, the performance of Prussian blue staining, a technique described by Max Perls in 1867, is needed. This method does not use a dye or colorant but uses hydrochloric acid to release the iron bound to proteins, and this later reacts with potassium ferrocyanide to form ferric ferrocyanide, an insoluble complex of iron with a characteristic blue-green color (Prussian blue). In sideroblastic anemias, iron retention in the macrophages is observed (in part due to intramedullary hemolysis) and the presence of ring sideroblasts (≥5 or hemosiderotic granules in a perinuclear position, covering at least one-third of the nuclear circumference) [10]. In congenital sideroblastic anemias, it is more common for ring sideroblasts to occur in late-stage erythroblasts, while in myelodysplastic syndromes, they are evident in all stages of erythroid maturation [51]. Figure 2 shows smears from a patient with MDS with ring sideroblasts.

## 7. Non-Clonal or Metabolic Acquired Sideroblastic Anemias

There are sideroblastic anemias attributed to certain medications or toxic exposure in which the anemia is fully reversible upon removal of the cause. The prevalence of these disorders is not well characterized. The most common causes of reversible sideroblastic anemia are described below.

### 7.1. Alcohol Consumption

Anemia in patients with excessive and chronic alcohol consumption is multifactorial, but it has been described that up to one-third of these patients might show ring sideroblasts [59]. They occur more frequently in patients with associated malabsorption. Alcohol and its metabolite, acetaldehyde, affect hemoglobin synthesis by exerting a reversible toxic effect on delta-aminolaevulinic acid synthetase [60]. An imbalance between the amount of iron imported into mitochondria and insufficient production of protoporphyrin IX to incorporate the iron may explain mitochondrial iron accumulation [60]. MCV is usually normal or elevated. In the peripheral blood smear, it is common to observe a double population of red blood cells with the presence of siderocytes. In bone marrow, erythroblast vacuolization and ring sideroblasts in terminal erythroblasts is a common feature. Ring sideroblasts presence could be reversible in days or weeks after stopping consumption. However, recovering from anemia may require a longer period of time, especially if alcohol consumption diminished folate reserves [61,62,63,64,65,66].

### 7.2. Drugs

The drugs most frequently associated with the development of ring sideroblasts are isoniazid and chloramphenicol, but others, such as linezolid, pyrazinamide, penicillamine, cycloserine, fusidic acid, melphalan, busulfan, and triethylenetetramine dihydrochloride, have also been reported.

#### 7.2.1. Isoniazid

Isoniazid is a hydrazide form of isonicotinic acid with antimycobacterial properties, and it is commonly used to treat tuberculosis in combination with other antimycobacterial agents or alone to prevent active infection in people in contact with the bacteria.

Two types of anemia associated with the consumption of isoniazid have been described: cases of pure red cell aplasia [67], characterized by acute normochromic normocytic anemia, reticulocytopenia, and bone marrow erythroblastopenia; and sideroblastic anemia, characterized by an increased erythropoiesis with ring sideroblasts in bone marrow. Both conditions are rare. It seems that there are predisposing situations for the development of sideroblastic anemia, such as concomitant folic acid deficiency. The mechanism of anemia appears to be related to the drug’s interference with B6 vitamin or pyridoxine metabolism, the main cofactor of the enzyme delta-aminolevulinic acid synthase (ALAS), resulting in a depletion of Heme synthesis [68,69]. Most patients have low B6 vitamin serum levels. The state is reversible a few weeks after withdrawing the drug or after starting supplementation with pyridoxine [70,71]. 

#### 7.2.2. Pyrazinamide

Pyrazinamide is a cornerstone antimicrobial agent that is commonly used for treatment during the initial phase of active tuberculosis. As with isoniazid, its use has been related to the presence of ring sideroblasts [72]. Pyrazinamide inhibits the enzyme 5-aminolevulinic acid synthase-2 (ALAS-2); therefore, iron accumulates within the mitochondrial matrix bound to mitochondrial ferritin, forming the ring sideroblasts [72].

#### 7.2.3. Chloramphenicol

The use of this drug has been more frequently associated with cases of aplastic anemia; however, cases of reversible microcytic and hypochromic anemia with ring sideroblasts have also been described. The drug alters the synthesis of mitochondrial proteins (similar to its bacteriostatic mechanism of action) [73,74]. Animal models have reported a decreased activity of ALAS and FECH activity in cases of sideroblastic anemia due to chloramphenicol intoxication [75]. 

Leiter et al. analyzed the effects of chloramphenicol on cellular iron metabolism based on the use of the K562 human erythroleukemia cell line [73]. Chloramphenicol decreased the activity of cytochrome c oxidase, reduced the ATP content of the cells, and inhibited oxidative metabolism. Chloramphenicol provoked ferrokinetic changes consisting of increased plasma iron, increased saturation of iron-binding globulin, delayed plasma clearance of Fe”, and decreased iron utilization [76]. 

#### 7.2.4. Linezolid

Linezolid is a synthetic oxazolidinone antimicrobial drug with bacteriostatic activity against Gram-positive organisms for Gram-positive infections and approved for the treatment of bacterial pneumonia, skin infections, and vancomycin-resistant enterococcal infections. Linezolid is known to have mitochondrial toxicity due to selective binding to mitochondrial ribosomes, inducing protein-synthesis inhibition [77,78]. Anemia and thrombocytopenia are well-reported adverse effects of linezolid [79,80]. This drug can cause vacuolization of immature erythroblasts, as well as the appearance of ring sideroblasts after a median exposure of 2 weeks. The mechanism of vacuole and ring sideroblasts formation may be mitochondrial injury [80,81]; however, further studies are needed to clarify its specific mechanism. 

### 7.3. Copper Deficiency

Copper is an essential element with a key role in several enzymatic reactions in RBCs. Ceruloplasmin is a ferroxidase that helps to convert ferrous to ferric iron, allowing it to bind transferrin and be transported throughout the body. The enzyme cytochrome oxidase, which is copper-dependent, is required for the reduction of ferric iron and incorporating it into the Heme molecule [82,83,84]. Copper deficiency can develop hematological abnormalities, and it might mimic myelodysplastic syndromes [85,86,87,88,89]. Patients with low copper levels can also present with neurological symptoms secondary to demyelination [90]. 

Copper deficiency can occur in different situations: malabsorption (intestinal or bariatric surgery), lack of supplementation in parenteral nutrition, excessive ingestion of zinc (induces the formation of a metalloprotein that sequesters copper at the intestinal level and prevents its absorption [82]), or chelation with dihydrochloride of triethylenetetramine.

The most common hematological abnormalities are anemia and neutropenia, while thrombocytopenia is rare [85,86,87,89,91]. Moreover, copper deficiency can also induce dysplastic features in hematopoietic precursors. Cytoplasmic vacuolization of myeloid and erythroid precursors, as well as ring sideroblasts, is commonly seen. Evidence of iron-containing plasma cells is another morphological finding described [85,92]. A hematogone increase by flow cytometry has also been detected in copper deficiency [93,94]. Ring sideroblasts presence usually reverts 2 months after correct supplementation; however, neurological symptoms may be irreversible.

### 7.4. Pyridoxine (Vitamin B6) Deficiency

Pyridoxal-5-phosphate, a metabolically active derivative of pyridoxine, serves as a coenzyme for a variety of reactions involving decarboxylation and transamination [95]. Pyridoxal-5-phosphate participates as a cofactor of 5-aminolevulinic acid synthase 2 (ALAS2) for the formation of delta-amino levulinic acid from glycine and succinyl-CoA, the first step in Heme synthesis. Pyridoxine deficiency is associated with the presence of ring sideroblasts and microcytic and hypochromic anemia [96,97]. 

### 7.5. Lead Intoxication

Lead poisoning can cause anemia and RBCs with basophilic stippling (which is one of the hallmarks of diagnosis) due to inhibition of pyrimidine 5-nucleotidase [98]. Anemia is usually microcytic and hypochromic; however, some studies describe a frequent association with coexistent iron deficiency or thalassemia trait in children with lead poisoning [99,100]. Lead intoxication has been recognized to cause sideroblastic anemia by inhibiting a wide variety of enzymes concerning Heme synthesis, such as coproporphyrin oxidase, δ-aminolevulinate dehydratase, and FECH [101]. Elevated free porphyrins may not be sufficient to produce ring sideroblasts since it can be observed in other situations as iron deficiency; therefore, myelodysplastic syndrome has to be excluded in adults with ring sideroblasts and lead poisoning.

### 7.6. Hypothermia

Some cases of reversible thrombocytopenia and sideroblastic anemia have been described in hypothermic patients [102]. The mechanism of action is not truly described, although it seems to be related to disturbance of mitochondrial metabolism and oxidative phosphorylation at low-temperature contexts [103]. Interestingly, anemia occurring in the context of hypothermia as a therapeutical approach in neonatal hypoxic injury has been reported [103,104,105,106]. 

## 8. Clonal Sideroblastic Anemias

Clonal conditions associated with ring sideroblasts include myeloid neoplasm such as MDS, myeloproliferative neoplasms, MDS/MPN overlap syndromes, and acute myeloid leukemia (AML). Among them, the presence of ring sideroblasts is a diagnostic criterion for two definite entities according to 2017 WHO classification: myelodysplastic syndromes with ring sideroblasts (MDS-RS) and myelodysplastic/myeloproliferative neoplasm with ring sideroblasts and thrombocytosis (MDS/MPN-RS-T) [7]. Since the presence of ring sideroblasts is an almost perfect surrogate for the presence of the *SF3B1* mutation, the latest WHO (fifth edition) classification and the International Consensus Classification (ICC) 2022 will replace the MDS-RS category with that of MDS with *SF3B1* mutation (MDS-SF3B1). In the next WHO 2022 classification MDS with low blasts and ring sideroblasts will be retained for describing those cases with wild-type *SF3B1* and ≥15% ring sideroblasts [107,108]. 

Table 2 shows diagnostic criteria for clonal sideroblastic anemias according to the 2017 WHO classification.

### 8.1. Myelodysplastic Syndromes with Ring Sideroblasts (MDS-RS)

#### 8.1.1. Definition

The most common acquired sideroblastic anemias are MDS-RS. According to 2017 WHO classification, MDS-RS is an MDS characterized by cytopenias, usually anemia, morphological dysplasia involving one or more myeloid lineages and ring sideroblasts representing ≥15% of the bone marrow erythroid precursors (or 5% in the presence of SF3B1 mutations). Myeloblasts account for <1% in peripheral blood and <5% in bone marrow, Auer rods are absent, and the diagnostic criteria for MDS with del(5q) isolated must be excluded. Secondary causes of ring sideroblasts must be ruled out [7].

The WHO classification considers two subtypes: cases with single-lineage dysplasia (MDS-RS-SLD), meaning patients with anemia and dysplasia limited to the erythroid lineage; and multilineage dysplasia (MDS-RSMLD), meaning patients with any cytopenia and features of dysplasia in at least two myeloid lineages [7,109,110,111].

#### 8.1.2. Epidemiology

MDS-RS-SLD constitutes approximately 3–10% of all MDS cases. Median age of presentation is around 60–73 years, with a minor male predominance. MDS-RS-MLD appears to be more frequent, comprising about 13% of all MDS with an age and gender distribution similar to MDS-RS-SLD [109,112,113].

#### 8.1.3. Clinical Features

The presenting symptom is usually anemia, most often macrocytic (MCV > 100 fL), whereas white blood cell and platelet counts are generally normal at presentation. Bicytopenia or pancytopenia occurs in a higher proportion of MDS-RS-MLD patients [7]. Most patients have evidence of iron overload, as indicated by increased serum iron, transferrin saturation, and serum ferritin. Ambaglio et al. observed an inappropriately low hepcidin levels in MDS-RS patients, which may result in excessive reticuloendothelial iron release and parenchymal iron loading, as occurs in congenital iron loading anemias due to ineffective erythropoiesis [114].

Organ damage by iron overload becomes clinically relevant in transfusion dependent patients. The anemia of MDS-RS is usually mild and stable but tends to exacerbate with time, eventually resulting in transfusion dependence [115]. A small proportion of patients may progress to AML [110,116,117].

#### 8.1.4. Microscopy

Peripheral blood smear may show an important anisocytosis and poikilocytosis with a double population of RBCs, a majority normochromic and a minority hypochromic. Siderocytes and RBCs with anomalous distribution of hemoglobin and basophilic stippling coexistence are usually observed. Blasts cells are absent or rare (accounting for <1% white blood cells). Cases with 1% PB blasts must be considered in the MDS unclassifiable (MDS-U) category since they appear to have more aggressive behavior [118].

In MDS-RS-SLD, smears of bone marrow aspirate stained with a panoptic stain show an increase in erythroid precursors with erythroid dysplasia as megaloblastoid changes, nuclear abnormalities, and basophilic stippling. Granulocytic and megakaryocytic dysplasia is not present or accounting for <10% dysplastic forms by definition. Blasts constitute <5% of the nucleated bone marrow cells. The Perls stain shows ≥15% ring sideroblasts (or ≥5% if the *SF3B1* mutation is present) of the erythroid precursors, defined as those erythroblasts with 5 iron granules in at least 1/3 of the nuclear contour [10]. In MDS-RS-MLD in addition to erythroid dysplasia and ring sideroblasts, there is significant dysplasia (≥10%) in one or two non-erythroid lineages.

The presence of ring sideroblasts can be seen in other subtypes of myelodysplastic syndrome. Patients with ring sideroblasts and excess of blasts in peripheral blood or bone marrow should be classified as MDS with excess blasts. Patients with ring sideroblasts and who fulfil MDS with isolated del(5q) criteria should be classified as such, even in the presence of the *SF3B1* mutation [7].

#### 8.1.5. Genetic and Molecular Profile

Clonal chromosomal aberrances are described in 5–20% of cases of MDS-RS-SLD, usually affecting a single chromosome, and in 50% of MDS-RS-MLD, more often presenting high-risk abnormalities [54,109,110,113]. There are no specific karyotypic alterations of these MDS; they usually present those aberrations observed in other MDS [119]. An association between primary MDS with del(11q) in a non-complex karyotype and the presence of ring sideroblasts has been found [120].

In 2011, two independent international cooperative groups performed whole-exome sequencing studies and identified the relationship between MDS-RS and somatically acquired mutations in *SF3B1*, a gene encoding a splicing factor, in a high proportion of patients [121,122]. As a consequence of these mutations, defects in Heme biosynthesis and iron accumulation in mitochondria are present [117]. *SF3B1* is a main component of the U2 snRP spliceosome that recognizes the 39 splice-acceptor sites in de novo transcribed mRNAs. A different set of hot spots of the *SF3B1* gene have been described, including codon 700, which affects 50% of cases, and less frequently, codons 666, 662, 622, and 625 [112,123].

Shiozawa et al. demonstrated three genes involved in iron metabolism and Heme biosynthesis that exhibited aberrant splicing in *SF3B1*-mutated patients: *ABCB7*, *PPOX*, and *TMEM14C* [124]. In human *SF3B1*-mutant MDS, ring sideroblasts are thought to arise as a result of aberrant splicing of key genes involved in Heme biosynthesis, such as *ABCB7*, *TMEM14C*, *ALAS2*, and *SLC25A37* [125,126,127,128]. The most frequent differentially spliced events in *SF3B1* mutated cases were alternative 3’ splice site [124,129]; this was also confirmed by Obeng et al. after generating a knock-in (KI) mice model with *SF3B1*^+/K700E^ [130]. Although the specific role of *SF3B1* mutations in MDS-RS is not curtained, downregulation of ABC, upregulation of ALAS2, and downregulation of ABCB7 have been reported in refractory anemia with ring sideroblasts (RARS), the term previously used for MDS-RS [131,132]. Ring sideroblasts were not observed in single-gene knock-out (KO) mice for either *ALAS2* [133] or *ABCB7* [134] did not recapitulate ring sideroblasts’ phenotype, suggesting that sideroblast formation may demand concurrent reduction in the levels of multiple proteins implicated in both Heme biosynthesis (TMEM14C, ALAS2) and mitochondrial iron transport (SLC25A37, ABCB7). Therefore, this phenotype may be difficult to reproduce in single-gene-targeting models. *SF3B1* is involved in cell growth, cell cycle, and erythroid differentiation [125,135]. Mupo et al, showed a reduction in mature erythroid cells in *SF3B1*^+/K700E^ animals [136]. Moreover, *SF3B1*^+/K700E^ results in a progressive macrocytic anemia [130]. Different studies have shown a decreased number and a compromised function of HSCs after *SF3B1* (KI)/(KO) relating *SF3B1* to hematopoiesis reconstitution [130,137].

*SF3B1* mutations have been described in approximately 70–90% of patients with MDS-RS-SLD and 75% of patients with MDS-RS-MLD [138]. The percentage of bone marrow ring sideroblasts often correlates with the *SF3B1* mutant variant allele frequency burden (VAF) [121,122,139,140]. Malcovati et al. studied the clinical significance of *SF3B1* mutations in MDS and MDS/MPN, reporting that the presence of *SF3B1* mutation has a positive predictive value for disease phenotype with ring sideroblasts of 97.7%, and the absence of this mutation has a negative predictive value of 97.8% [141]. Subsequently, *SF3B1* is the first gene deeply related to a specific morphology in myeloid malignancies. *SF3B1* mutations have an impact on the phenotype of MDS and are clearly associated to dysregulated expression of genes involved in mitochondrial and iron metabolism.

Recently, the International Working Group for the Prognosis of MDS (IWG-PM) has proposed a modification in the classification of MDS, not defining a subgroup based on the presence of ring sideroblasts, but rather on the presence of the *SF3B1* mutation. They define *SF3B1*-mutant MDS with the following criteria [138]:-Cytopenia as defined by standard hematologic values,-Somatic *SF3B1* mutation,-Morphologic dysplasia (erythroid or multilineage dysplasia), with or without ring sideroblasts,-Bone marrow blasts <5% and peripheral blood blasts <1%,-Not meet WHO criteria for MDS with isolated del(5q), MDS/MPN-RS-T or other MDS/MPNs, and primary myelofibrosis or other MPNs,-Normal karyotype or any cytogenetic abnormality other than del(5q); monosomy 7; and inv(3) or abnormal 3q26, complex (≥3),-Any additional somatically mutated gene other than *RUNX1* and/or *EZH2* can be present.

The authors concluded that *SF3B1*-mutant MDS represents a distinct entity, mainly characterized by ineffective erythropoiesis, relatively good prognosis, and potential response of anemia to luspatercept treatment [142,143,144].

An entity of MDS with mutated SF3B1 (MDS-*SF3B1*) has been included in the two new classifications of myeloid neoplasms recently published: the World Health Organization (WHO) 2022 classification and the International Consensus Classification (ICC) 2022 [107,108]. In the ICC, the diagnostic criteria are the same as those described in the IWG proposal, except that the presence of the *EZH2* mutation does not invalidate the diagnosis of this entity.

In addition to *SF3B1*, another association between a gene defect and the ring sideroblasts phenotype was defined for *PRPF8*, for which mutations are reported in approximately 3% of myeloid neoplasms, including MDS, MDS/MPN, and secondary AML [145]. *SRSF2* and *ZRSR2* mutations are also observed in *SF3B1*-unmutated MDS-RS mutations; however, it requires further studies to establish if these mutations are enriched in MDS-RS when compared to the MDS population [122,145].

Recently, Swoboda et al. studied bone marrow ring sideroblast percentage and its correlation with *TP53* mutational state in patients diagnosed of MDS with excess blasts [146]. After analyzing 218 patients with MDS with ring sideroblasts ≥5%, investigators suggested *SF3B1*-unmutated MDS-RS as a distinct entity to the mutant counterpart characterized by increased prevalence of MDS with excess blasts, complex karyotype, lower peripheral blood counts, and *TP53* mutation.

#### 8.1.6. Prognosis

Patients with MDS-RS are usually stratified into lower-risk categories by using classical MDS prognostic scores, systems such as the IPSS (international prognostic scoring system), the revised IPSS (R-IPSS), the Low-Risk Prognostic Scoring System (LR-PSS), and the WPSS (WHO classification based prognostic scoring system) [111,147,148,149].

The median overall survival (mOS) for patients with MDS-RS-SLD ranges from 69 to 108 months, with a very low risk for leukemic transformation (<2%). In MDS-RS-MLD, mOS is approximately 28 months, and around 8% of patients progress to AML [112,116,150].

Li et al. studied 230 consecutive MDS patients with the presence of at least 1% ring sideroblasts and without excess blasts [151]. Interestingly, no significant difference in survival was observed among patients with 5–15% ring sideroblasts and *SF3B1* mutations and individuals with 15% ring sideroblasts, regardless of the SF3B1 mutation status. However, patients with 5–15% ring sideroblasts with *SF3B1* mutations showed better overall survival compared to those without. This was the foundation for the 2017 WHO classification to consider MDS with 5–15% ring sideroblasts and *SF3B1* mutation within MDS-RS category [7].

Patnaik et al. defined the prognostic irrelevance of BM ring sideroblasts percentage after analyzing 200 patients with MDS without excess blasts and ≥1% ring sideroblasts and assessing the impact of ring sideroblasts % as both a continuous and categorical variable [152]. 

The prognostic impact of *SF3B1* mutations in MDS-RS has been controversial. Some reports have demonstrated a favorable independent prognostic impact [141], while others did not confirm this [112,153,154]. The International Working Group for the Prognosis of MDS assessed the impact of *SF3B1* mutations in 3749 MDS patients (795 *SF3B1*-mutated) and concluded that *SF3B1*-mutated MDS represented a unique MDS subtype with favorable outcomes regardless of the presence of ring sideroblasts [138].

Several studies have revealed that single- or multilineage dysplasia according to the WHO morphological criteria do not show effect on survival or risk of disease progression within *SF3B1*-mutated patients [117,138]. 

A new prognostic index for MDS has recently been published, the molecular IPSS-R [155]. *SF3B1* mutations were associated with favorable outcomes; however, this association was strongly modulated by patterns of commutation. A cluster analysis segregated *SF3B1*-mutated cases into three independent groups:-SF3B1^5q^ for concomitant presence with isolated del(5q) (7% of *SF3B1*-mutant);-SF3B^b^ as the commutation between *SF3B1* and any gene from *BCOR*, *BCORL1*, *NRAS*, *RUNX1*, *SRSF2*, or *STAG2* (15% of SF3B1-mutant);-SF3B1^a^ as any other mutant *SF3B1* (78% of *SF3B1*-mutant), with 107 patients (19%).

The favorable outcomes associated with *SF3B1* mutations were confined to the *SF3B*^a^ group and not observed for *SF3B1*^5q^ or *SF3B1*^b^, including in low blast disease [155].

#### 8.1.7. Treatment

Supportive treatment methods for individuals with MDS-RS [156] include RBC and platelet transfusions; erythropoiesis-stimulating agents (ESA) [157,158]; immunomodulatory agents, such as lenalidomide [159,160]; TGF-β superfamily members’ regulators (Sotatercept [161] and Luspatercept [144]) and iron chelation therapy [162,163]; hypomethylating agent therapy [164] (Azacitidine [165], oral azacitidine [166], Decitabine [167], or combination of decitabine with cedazuridine [168]); and telomerase inhibitors, such as imetelstat [169], among others. New drugs targeting specific splicing modulators are currently undergoing clinical trials, aiming to make synthetic lethality a new therapeutical approach [170]. 

Allogeneic stem-cell transplantation remains as the only curative strategy for patients with MDS [156], despite the fact that, depending on individual risk factors, treatment-related mortality ranges from 15 to 50% [171,172].

### 8.2. Myelodysplastic Syndrome/Myeloproliferative Neoplasm with Ring Sideroblasts and Thrombocytosis (MDS/MPN-RS-T)

#### 8.2.1. Definition

Myelodysplastic syndrome/myeloproliferative neoplasm with ring sideroblasts and thrombocytosis (MDS/MPN-RS-T) is a new entity in the 2017 WHO classification. It is characterized by anemia with erythroid dysplasia, with or without multilineage dysplasia; thrombocytosis (platelet count ≥450 × 10^9^/L); and bone marrow ring sideroblasts ≥15%.

The WHO defined it as a provisional entity in 2001 as part of the MDS/MPN category [173]. In 2008, the revised WHO classification lowered the platelet threshold for diagnosis of MDS-RS-T from 600 × 10^9^/L to 450 × 10^9^/L, in line with the change in diagnostic criteria for essential thrombocythemia (ET), a myeloproliferative neoplasm characterized by the presence of thrombocytosis [11].

The current WHO diagnostic criteria for MDS/MPN-RS-T include the presence anemia; dyserythropoiesis in the bone marrow, with ring sideroblasts accounting for 15% or more of erythroid precursors, thrombocytosis (≥450 × 10^9^/L); and proliferation of large and morphologically atypical megakaryocytes similar to those of essential thrombocythemia (ET), showing enlarged, mature megakaryocytes with hyperlobulated nuclei. The peripheral blood blast cells should be <1%, and bone marrow blasts should be <5%. The absence of *BCR-ABL1*, *PDGRA*, *PDGFRB*, *FGR1*, and *PCM1-JAK2* rearrangements, as well as absence of t(3;3) (q21q26), inv(3) (q21q26), or del (5q), is also a diagnostic requirement. Additional criteria for the diagnosis of MDS/MPN-RS-T include the presence of *SF3B1* mutations with ≥15% ring sideroblasts and no prior history of MDS or MPN, with the exception of MDS-RS [11]. MDS/MPN with ring sideroblasts and thrombocytosis have been redefined in the 2022 WHO classification based on *SF3B1* mutation and renamed MDS/MPN with *SF3B1* mutation and thrombocytosis [107]. 

#### 8.2.2. Epidemiology

The overall mean age for presentation commonly ranges from 71 to 75 years, higher than myeloproliferative neoplasms age of diagnosis. Several series have described a slight predominance in women [174,175,176].

#### 8.2.3. Clinical Features

The clinical features of MDS/MPN-RS-T are similar to those observed in ET; however, anemia, frequently macrocytic, is always present. At the time of diagnosis, MDS/MPN-RS-T patients usually experience higher hemoglobin levels, WBC counts, and platelet levels than MDS-RS patients, but the levels are lower than those of ET patients [177].

#### 8.2.4. Microscopy

Peripheral blood smear may show anisocytosis, with a dimorphic pattern. RBCs with anomalous distribution of hemoglobin and basophilic stippling coexist are usually observed. Blasts cells are exceptional, accounting for <1% white blood cells. The platelets often show anisocytosis, with some large and atypical forms. The WBC count and leukocyte differential count are usually within normal ranges. Smears of bone marrow aspirate that are stained with a panoptic stain show an increase in erythroid precursors with erythroid dysplasia as megaloblastoid changes, nuclear abnormalities, and basophilic stippling. Multilineage dysplasia may be seen in some cases. Megakaryocytes are increase in number and morphologically atypical, similar to those of ET. The Perls’ stain shows ≥15% ring sideroblasts of the erythroid precursors, defined as those erythroblasts with five iron granules in at least 1/3 of the nuclear contour [7,174,177].

#### 8.2.5. Genetic and Molecular Profile

*SF3B1* mutations can be observed in approximately 85% of patients with MDS/MPN-RS-T. Moreover, as it occurs in ET, mutations in *JAK2*, particularly the V617F hotspot mutation, have been depicted in up to 50% of patients. About half of the patients harbor both the *JAK2* V617F and the *SF3B1* mutations. Mutations in other genes of the splicing machinery (*SRSF2*, *U2AF1*, and *ZRSR2*) as well as in signaling pathway genes (*CBL*, *MPL*) have been described. Likewise, other mutations frequently observed in myeloid neoplasms (*ASXL1*, *DNMT3A*, *TET2*, *ETV6*, *RUNX1* or *SETBP1*) have been observed in these patients [174,175,176]. Unlike in ET, MPL and *CALR* are infrequent in MDS/MPN-RS-T [178].

Regarding chromosome analysis, clonal cytogenetic changes can be seen in 5–20% of patients, including del (7q) (q12q21), del (11) (q23), +8, −Y, −7, and del12p (1.1–1.3) [179].

#### 8.2.6. Prognosis

In a large retrospective study including 200 patients diagnosed with MDS-RS, MDS/MPN-RS-T, and ET, the three conditions were compared. MDS/MPN-RS-T patients showed higher mOS than MDS-RS-SLD (76 months vs. 63 months), but a lower mOS than ET (76 months vs. 117 months) [177].

A recent Mayo-Moffitt collaborative study of 158 patients with MDS/MPN-RS-T investigated their clinical and prognostic features and compared them with MDS/MPN-U [180]. In a multivariate analysis, only abnormal karyotype and hemoglobin ≤10 g/dL independently predicted shorter survival.

#### 8.2.7. Treatment

Due to its novelty definition, the current therapeutic guidelines were developed from related myeloid neoplasms, such as MDS-RS and MPN with a low risk (ET) [156]. Patients with anemia are treated similarly to those with lower-risk MDS, with early administration of ESA and transfusional supportive treatment [181]. Antelo et al. analyzed 44 patients with MDS/MPN-RS-T who receive ESA at any time since diagnosis, where erythroid response was observed in 45% of the patients, with median duration of response being 20 months [181]. BM ring sideroblasts’ percentage did not impact ESA response. Different drugs have been proposed as cornerstones of MDS/MPN-RS-T: lenalidomide [182,183], luspatercept [144], antiplatelet agent [184], and cytoreductive therapy.

### 8.3. Other Myeloid Neoplasms with Ring Sideroblasts

#### 8.3.1. Primary Myelofibrosis

Lasho et al. particularly described the clinicopathological features associated with *SF3B1* mutation in primary myelofibrosis [185]. *SF3B1* mutations were detected in 6.5% of the patients (10/155), of which six patients also exhibited ring sideroblasts. The authors demonstrated the genuine occurrence of *SF3B1* mutations in PMF and their apparently invariable association with high percentage of bone marrow ring sideroblasts. No associations between the presence of *SF3B1* mutations and clinical features were detected, with the exception of marked splenomegaly that occurred more often in *SF3B1*-mutated cases. Leonardo et al. studied features of *SF3B1*-mutated myeloproliferative neoplasms (MPNs) [186]. Of note, ring sideroblasts were present only in a subset of *SF3B1*-mutated cases (4 out of 10) with no other features of erythroid dysplasia. Other case reports cases have supported these suggestions [140,187,188].

#### 8.3.2. Acute Myeloid Leukemia 

Ring sideroblasts can also be present in a subset of patients with acute myeloid leukemia (AML), ranging from 5 to 16% [141,189], whereas *SF3B1* mutations are infrequent [141,190]. Berger et al. studied 126 AML patients with ring sideroblasts ≥1% [191]. AML-RS subjects were enriched in the ELN adverse risk category with 35% of all cases showing cytogenetic aberrancies. Among this cohort, a gene panel using NGS was performed in a subset of 60 patients where *TP53* was most recurrently mutated in this cohort (37%), followed by *DNMT3A* (26%), *RUNX1* (25%), *TET2* (20%), and *ASXL1* (19%). *TP53* mutation was especially detected in patients with higher ring sideroblasts percentages (73% in the ≥15% ring sideroblasts group vs. 12% in the 1–4% ring sideroblasts group). AML-RS and MDS-RS may present comparable downstream effector pathways, particularly regarding to Heme metabolism. However, their genetic backgrounds clearly diverge.

Martin-Cabrera et al. presented an interesting genomic analysis of 340 patients with AML with ring sideroblasts (89% de novo AML). Bone marrow blasts inversely correlated with the percentage of ring sideroblasts. This is possibly connected to the higher occurrence of M2 and M4 subtypes, happening to be the most prevalent categories happening after MDS-RS. The authors described a molecular signature (*ASXL1*mut and *SF3B1*mut) previously defined in s-AML [192], suggesting a possible MDS background among de novo AML with ring sideroblasts. 

## 9. Discussion

Although sideroblastic anemia remains as an uncommon entity, it should be considered in all children/infants and adults with unexplained anemia. The key to its diagnosis is evidencing ring sideroblasts in the bone marrow. Thanks to the advances in genomics in recent years, a wide spectrum of causes of sideroblastic anemia, both hereditary and acquired, has been described. Nevertheless, our understanding of the downstream events that lead to ring sideroblast formation and anemia is limited. The distinction between the different forms of sideroblastic anemia is helpful for predicting the disease course and guiding therapy.

## Figures and Tables

**Figure 1 genes-13-01562-f001:**
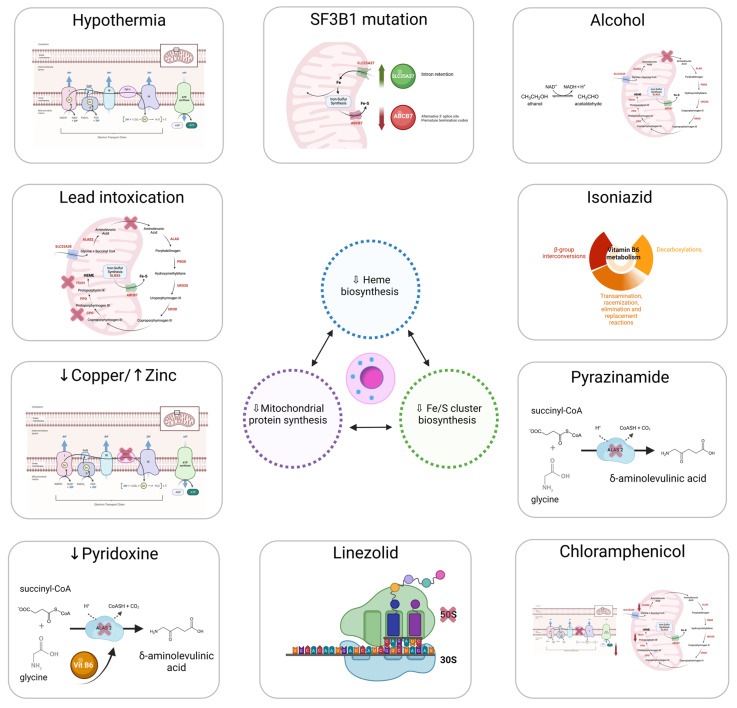
Main causes of acquired sideroblastic anemia.

**Figure 2 genes-13-01562-f002:**
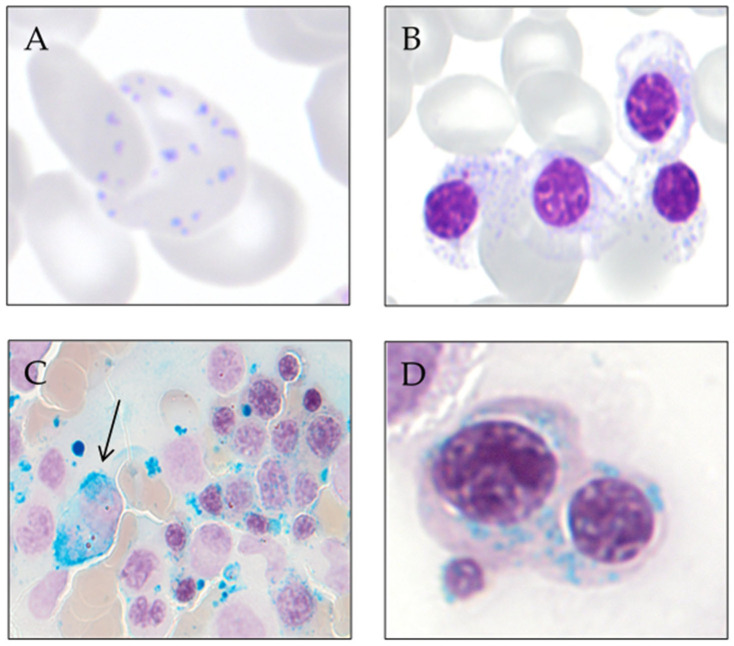
Smears from a patient with MDS with ring sideroblasts. (**A**) Peripheral blood red cell with coexistence of anomalous distribution of hemoglobin and basophilic stippling (May-Grünwald Giemsa). (**B**) Bone marrow erythroblast with poorly hemoglobinized cytoplasm and basophilic stippling (May-Grünwald Giemsa). (**C**) Bone marrow smear at low magnification showing an iron-laden macrophage (arrow) and numerous ring sideroblasts (Perls’ reaction). (**D**) Bone marrow ring sideroblasts (Perls’ reaction).

**Table 1 genes-13-01562-t001:** Genetic and phenotypic characteristics of sideroblastic anemias (Adapted from Ducamp et al. [1]).

	Inheritance	Gene	Syndromic	Age at Presentation	Anemia Severity	MCV	Other Symptoms
Congenital
Heme synthesis defects
XLSA	X	*ALAS2* (* 301300)	No	Infancy to adulthood	Mild to severe	↓ 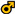 N/↑ 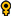	Iron overload in the absence of transfusions
SLC25A38	AR	*SLC25A38 (* 610819)*	No	Infancy	Severe	↓	Transfusional iron overload
Erythropoietic protoporphyria	AR/PSD	*FECH (* 612386)*	No	Childhood	Mild	↓	Acute photosensitivity
Fe-S biogenesis defects
GLRX5 deficiency	AR	*GLRX5* *(* 609588)*	No	Adulthood	Mild to severe	↓	Iron overload
HSPA9 deficiency	AR/PSD	*HSPA9* *(* 600548)*	No	Childhood	Mild to severe	N/↓	Retinitis pigmentosa
HSCB deficiency	AR	*HSCB* *(* 608142)*	No	Childhood	Moderate	N	None
XLSA/A	X	*ABCB7* *(* 300135)*	Yes	Childhood	Mild to moderate	↓	Cerebellar ataxia and hypoplasia, delayed motor development
Mitochondrial protein synthesis defects
PMPS	SP/M	mtDNA	Yes	Infancy	Severe	↑	Lactic acidosis, exocrine pancreatic insufficiency, failure to thrive, hepatic/renal failure
MLASA1	AR	*PUS1* *(* 608109)*	Yes	Childhood	Mild to severe	N/↑	Myopathy, lactic acidosis, facial dysmorphism
MLASA2	AR	*YARS2* *(* 610957)*	Yes	Childhood	Mild to severe	N/↑	Myopathy, lactic acidosis, cardiomyopathy
LARS2 deficiency	AR	*LARS2* *(* 604544)*	Yes	Infancy	Severe	↑	Lactic acidosis, cardiomyopathy,hepatopathy, seizures
SIFD	AR	*TRNT1* *(* 612907)*	Yes	Infancy	Severe	↓	Immunodeficiency,aseptic febrile episodes,developmental delay,seizures, cardiomyopathy,retinitis pigmentosa, other
Mitochondrial protein synthesis defects
MT-ATP6-SA	SP/M	*MT-ATP6**(** 516060)	Yes	Infancy to early childhood	Mild to severe	N/↑	Lactic acidosis, myopathy,neurological abnormalities
NDUFB11-SA multifactorial	X	*NDUFB11 (* 300403)*	Yes	Early childhood	Moderate	N	Lactic acidosis, myopathy
TRMA	AR	*SLC19A2 (* 603941)*	Yes	Early childhood	Mild to severe	↑	Sensorineural deafness, non-type-I diabetes mellitus, optic atrophy, stroke-like episodes
Acquired
MDS-RS-SLD	Somatic	*SF3B1* *(* 605590)*	N/A	Adulthood	Mild to moderate	↑/N	Iron overload
MDS-RS-MLD	Somatic	*SF3B1*	N/A	Adulthood	Mild to moderate	↑/N	Iron overload, other cytopenias
MDS/MPN-RS-T	Somatic	*SF3B1**JAK2**(* 147796)*, *CALR**(* 109091)* o *MPL**(* 159530)*	N/A	Adulthood	Mild	↑/N	Thrombocytosis

Abbreviations: ↓, decreased; ↑, increased; AR, autosomal recessive; M, maternal; MCV, mean red blood cell volume; MDS/MPN, myelodysplastic syndrome/myeloproliferative neoplasm; MDS/MPN-RS-T, MDS/MPN with ring sideroblasts and thrombocytosis; MDS-RS-MLD, MDS with ring sideroblasts and multilineage dysplasia; MDS-RS-SLD, MDS with ring sideroblasts and single-lineage dysplasia; N, normal; N/A, not applicable; PMPS, Pearson marrow–pancreas syndrome; SIFD, SA, immunodeficiency, fevers, and developmental delay; SP, sporadic; TRMA, thiamine-responsive megaloblastic anemia; X, X-linked; XLSA, X-linked SA; XLSA/A, X-linked CSA associated with cerebellar ataxia.

**Table 2 genes-13-01562-t002:** Diagnostic criteria for clonal sideroblastic anemia (adapted from Swerdlow et al. [7]).

	Cytopenia/s	Dysplastic Lineages	Blasts	% RS	Others
MDS-RS-SLD	1 or 2	1	<1% PB and <5% BM	≥15 or ≥5 if *SF3B1 mut*	No MDS 5q criteria No Auer rods
MDS-RS-MLD	1–3	≥2	<1% PB and 5% BM	≥15 or ≥5 if *SF3B1 mut*	No MDS 5q criteria No Auer rods
MDS/MPN-RS-T	Anemia	1–3	<1% PB and 5% BM	≥15	Thrombocytosis

Abbreviations: RS, ring sideroblasts; MDS, myelodysplastic syndromes; SLD, single-lineage dysplasia; MLD, multiple lineage dysplasia; MDS/MPN-RS-T, myelodysplastic syndrome/myeloproliferative neoplasm with ring sideroblasts and thrombocytosis.

## Data Availability

Not applicable.

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
