# Peer review of "Causes and Pathophysiology of Acquired Sideroblastic Anemia"

_genes, 2022, doi:10.3390/genes13091562_

Round 1

Reviewer 1 Report

This is a thorough review article on the understanding of acquired sideroblastic anemia. While the manuscript is informative, this reviewer considers that additional re-organization would be required in this form of the manuscript.

Specific comments

1.     There are several repetitions, such as ‘6. Molecular landscape in myeloid neoplasms with ring sideroblasts (page 8)’ and ‘5) genetic and molecular profile (page 14)’. Thus, re-organization would be required.

2.     In the section of ‘5. Ring sideroblast formation’, it would be informative to discuss the supposed mechanism of ring sideroblast formation in each defect of causative gene for congenital sideroblastic anemia. 

3.     There are several typos, which should be corrected.

Author Response

This is a thorough review article on the understanding of acquired sideroblastic anemia. While the manuscript is informative, this reviewer considers that additional re- organization would be required in this form of the manuscript.

Specific comments

  1. There are several repetitions, such as ‘6. Molecular landscape in myeloid neoplasms with ring sideroblasts (page 8)’ and ‘5) genetic and molecular profile (page 14)’. Thus, re-organization would be required.

Thank you very much for your comment. We agree with the reviewer's remark, so we have revised these sections and reorganized the text so that there are no redundant or repetitive segments and the text is more pleasant for the reader to follow. Specifically, we have eliminated section 6 and included part of its information in section 8 (8.1. Myelodysplastic syndromes with ring sideroblasts (MDS-RS). 8.1.5. Genetic and molecular profile).

  1. In the section of ‘5. Ring sideroblast formation’, it would be informative to discuss the supposed mechanism of ring sideroblast formation in each defect of causative gene for congenital sideroblastic anemia. 

Thank you very much for your suggestion. Although we agree with the reviewer that it would be interesting to discuss the proposed mechanisms according to the condition of each gene in congenital sideroblastic anemia, we believe that it is slightly outside the scope of the review, since it is focused on Causes And Pathophysiology of Acquired Sideroblastic Anemia.

  1. There are several typos, which should be corrected.

Thank you very much for your comment. We have revised the text in depth together with an English-native speaking editor proofreading. We hope to have corrected these errors and to have improved the overall quality of the manuscript.

Reviewer 2 Report

This is a very useful state of the art review of “acquired sideroblastic anemias”.  Sections 1 thru 8 are strong. Section 9 is good as well until section 9.2.  I believe section 9.2 is a separate chapter in itself.

Specific points

1)      Figure 1 is critical but the details are difficult read as the text is out of focus even in the separate PNG file- the figure as it is now  detracts from the over all merit of the review

2)      This chapter is focused on acquired sideroblastic anemias but the whole field started with the paper by Thomas Cooley  describing a sex linked anemia (Cooley TB: A severe type of hereditary anemia with elliptocytosis. Interesting sequence of splenectomy. Am J Med Sci 209561, 1945) with later identification of mutations in ALAS2 and ringed sideroblasts by Cotter et al in the same family ( Blood 1994 Dec 1;84(11):3915-24. ) . It would be useful to mention this historical detail in section 2.

3)      Suggest including MIM numbers for all of the genes mentioned in this chapter

4)      P 9 Suggest including a schematic of the Fenton Reaction

5)      Lead poisoning: In my opinion lead poisoning causing microcytic anemia is week- two article show that in children with lead poisoning, the microcytic anemia is almost always due to either concurrent iron deficiency or thalassemia trait (Am J Dis Child 1990 Nov;144(11):1231-3.  doi: 10.1001/archpedi.1990.0215035006302 ; Pediatrics1988 Feb;81(2):247-54. PMID3277157.  Elevated free porphyrins per se may not be sufficient as it is elevated in iron deficiency and MDS has to be excluded in adults with lead poisoning

6)      Hypothermia as a cause of sideroblastic anemia is interesting- hypothermia is used as treatment to prevent neonatal hypoxic injury. Any references relevant to anemia in neonates receiving such therapy would be  useful

Author Response

This is a very useful state-of-the-art review of “acquired sideroblastic anemias”.  Sections 1 thru 8 are strong. Section 9 is good as well until section 9.2.  I believe section 9.2 is a separate chapter in itself.

Specific points

  • Figure 1 is critical but the details are difficult read as the text is out of focus even in the separate PNG file- the figure as it is nowdetracts from the over all merit of the review.

Thank you very much for your comment. We have again edited this image with the highest quality allowed by the web application biorender. We hope that after the changes made, the figure is to your liking.

  • This chapter is focused on acquired sideroblastic anemias but the whole field started with the paper by Thomas Cooley  describing a sex linked anemia (Cooley TB: A severe type of hereditary anemia with elliptocytosis. Interesting sequence of splenectomy. Am J Med Sci 209561, 1945) with later identification of mutations in ALAS2 and ringed sideroblasts by Cotter et al in the same family ( Blood 1994 Dec 1;84(11):3915-24.) . It would be useful to mention this historical detail in section 2.

Thanks to the reviewer for the comment, we have incorporated this interesting information into the historical context section and added both references (Lines 47-50).

In 1945, Cooley3 described a patient with sex-linked anemia which probably corresponded to a case of the non-syndromic form of X-linked SA (XLSA), since later Cotter et al identified mutations in ALAS2 and ringed sideroblasts in the same family4

3)      Suggest including MIM numbers for all of the genes mentioned in this chapter.

Thank you very much for your comment. According to the reviewer's suggestion, we have added the OMIM code to the different genes mentioned in the manuscript.

4)      P 9 Suggest including a schematic of the Fenton Reaction.

Thank you very much for your comment. We have designed a schematic figure representing the Fenton reaction.

Figure 3. The Fenton reaction. The Fenton reaction involves iron II (Fe2+) reacting with H2O 2 to yield a hydroxy radical (OH.) and a hydroxide ion (OH-).

5)      Lead poisoning: In my opinion lead poisoning causing microcytic anemia is week- two article show that in children with lead poisoning, the microcytic anemia is almost always due to either concurrent iron deficiency or thalassemia trait (Am J Dis Child 1990 Nov;144(11):1231-3.  doi: 10.1001/archpedi.1990.0215035006302 ; Pediatrics1988 Feb;81(2):247-54. PMID3277157.  Elevated free porphyrins per se may not be sufficient as it is elevated in iron deficiency and MDS has to be excluded in adults with lead poisoning.

Thanks to the reviewer for such an interesting remark. We have included this information in the text (lines 408-411) as well as adding the references proposed by the reviewer.

“Anemia is usually microcytic and hypochromic; however some studies describe a frequent association with coexistent iron deficiency or thalassemia trait in children with lead poisoning”

“Elevated free porphyrins may not be sufficient to produce ring sideroblasts since it can be observed in other situations as iron deficiency, therefore myelodysplastic syndrome has to be excluded in adults with ring sideroblasts and lead poisoning”. 

6)      Hypothermia as a cause of sideroblastic anemia is interesting- hypothermia is used as treatment to prevent neonatal hypoxic injury. Any references relevant to anemia in neonates receiving such therapy would be useful.

Thank you very much for your comment. In order to adequately respond to the reviewer, we have reviewed the literature regarding the adverse effects associated with hypothermia as an approach to prevent neonatal hypoxic injury.

In 1982, O'Brien et al. presented 3 patients in whom hypothermia treatment induced anemia (data regarding sideroblasts not available) that reversed in 2 of them as the body temperature returned to normal. Conversely, Lemyre et al. proposed anemia as an outcome of infants with hypoxic-ischemic encephalopathy whether they receive therapeutic hypothermia or not. Moreover, Dorothea J. Eicher et al. analysed the outcomes and the safety profile of hypothermia as an approach in neonatal encephalopathy and compared the same with respect to normothermia. After enrolling total of 32 hypothermia and 33 normothermia neonates, anemia was more frequent in the first group (14 vs 8, p= 0.093) although it did not reach statistical significance, probably due to a relatively small sample.

After this extensive literature search we have added the following text (lines 416-418):

“Interestingly, anemia occurring in the context of hypothermia as therapeutical approach in neonatal hypoxic injury has been reported1-4

References

  1. O'Brien, H.; Amess, J. A.; Mollin, D. L., Recurrent thrombocytopenia, erythroid hypoplasia and sideroblastic anaemia associated with hypothermia. British journal of haematology 1982, 51 (3), 451-6.
  2. Eicher, D. J.; Wagner, C. L.; Katikaneni, L. P.; Hulsey, T. C.; Bass, W. T.; Kaufman, D. A.; Horgan, M. J.; Languani, S.; Bhatia, J. J.; Givelichian, L. M.; Sankaran, K.; Yager, J. Y., Moderate hypothermia in neonatal encephalopathy: safety outcomes. Pediatric neurology 2005, 32 (1), 18-24.
  3. Gluckman, P. D.; Wyatt, J. S.; Azzopardi, D.; Ballard, R.; Edwards, A. D.; Ferriero, D. M.; Polin, R. A.; Robertson, C. M.; Thoresen, M.; Whitelaw, A.; Gunn, A. J., Selective head cooling with mild systemic hypothermia after neonatal encephalopathy: multicentre randomised trial. Lancet (London, England) 2005, 365 (9460), 663-70.
  4. Lemyre, B.; Chau, V., Hypothermia for newborns with hypoxic-ischemic encephalopathy. Paediatrics & child health 2018, 23 (4), 285-291.

Round 2

Reviewer 2 Report

The revisions address the prior review and the authors include an appropriate caution to exclude MDS in adults with ringed sideroblasts and lead poisoning- perhaps even copper/Zinc/pyridoxine deficiency should be excluded in such cases. The authors included new references in the text but these are not added to the reference list at the end of the chapter and thus the references are not in sync- e,g The Cooley ref (#2), Cotter ref (# 3), the two references re concurrent Iron deficiency and thal trait  in the text (# 92/93- Bhambhani; Aronow AJDC. 1990;144:1231-1233; Clark, Royal, Seeler Pediatrics1988 Feb;81(2):247-54.) are not listed in the references. Like wise the references to hypothermia also appear to be misaligned.

Fig 1 is critical and suggest making sure the text is focused and sharpened.

Author Response

Revisions Round 2

Comments and Suggestions for Authors

Reviewer 2

The revisions address the prior review and the authors include an appropriate caution to exclude MDS in adults with ringed sideroblasts and lead poisoning- perhaps even copper/Zinc/pyridoxine deficiency should be excluded in such cases. The authors included new references in the text but these are not added to the reference list at the end of the chapter and thus the references are not in sync- e,g The Cooley ref (#2), Cotter ref (# 3), the two references re concurrent Iron deficiency and thal trait  in the text (# 92/93- Bhambhani; Aronow AJDC. 1990;144:1231-1233; Clark, Royal, Seeler Pediatrics. 1988 Feb;81(2):247-54.) are not listed in the references. Like wise the references to hypothermia also appear to be misaligned.

We thank the reviewer for their contributions, we think that the article has improved by incorporating them.

In revision 1 we provided a document that, as the reviewer points out, did not have the references refreshed. We attach the document correcting this error.

Fig 1 is critical and suggest making sure the text is focused and sharpened.

We have again edited this image with the highest quality allowed by the web application biorender. We hope that after the changes made, the figure is to your liking.
